# Invasive Meningococcal Disease in Children: Outcomes and Risk Factors for Sequelae and Fatal Cases in Greece

**DOI:** 10.3390/microorganisms13040705

**Published:** 2025-03-21

**Authors:** Panagiotis Poulikakos, Dimitrios Kapnisis, Athanasia Xirogianni, Irini Liakou, Maria Tsolia, Athanasios Michos, Elpis Mantadakis, Vassiliki Papaevangelou, Andreas Iliadis, Despoina Gkentzi, Stavroula Kostaridou Nikolopoulou, Maria Sdougka, Konstantina Charisi, Athanasios Bangeas, Evangelia Farmaki, Georgina Tzanakaki

**Affiliations:** 1National Meningitis Reference Laboratory, Surveillance Laboratory of Infectious Diseases, Department Public Health Policy, School of Public Health, University of West Attica, 115 21 Athens, Greece; ppoulikakos@uniwa.gr (P.P.); mdy22108@uniwa.gr (I.L.); 2Intensive Care Unit, Anticancer Hospital “Metaxa”, 185 37 Pireaus, Greece; 3Department of Pediatrics, Patras Medical School, University General Hospital of Patras, 265 04 Rio, Greece; kapn.dim@hotmail.com (D.K.); gkentzid@gmail.com (D.G.); 4Second Department of Paediatrics, National and Kapodistrian University of Athens, P & A Kyriakou Children’s Hospital, 115 27 Athens, Greece; mariantsolia@gmail.com; 5First Department of Pediatrics, Infectious Diseases and Chemotherapy Research Laboratory, National and Kapodistrian University of Athens, ‘Aghia Sophia’ Children’s Hospital, 115 27 Athens, Greece; amichos@med.uoa.gr; 6University General Hospital of Alexandroupolis, Democritus University, Alexandroupolis, 691 00 Thrace, Greece; emantada@med.duth.gr; 7Third Department of Pediatrics, National and Kapodistrian University of Athens, University General Hospital “ATTIKON”, 124 62 Athens, Greece; vpapaev@gmail.com; 8Pediatric Intensive Care Unit, University General Hospital of Patras, 265 04 Rio, Greece; picugreece@gmail.com; 9Department of Pediatrics, “Pentelis” Children’s Hospital, 152 36 Athens, Greece; vkostaridou@gmail.com; 10Pediatric Intensive Care Unit, Hippokration General Hospital, 546 42 Thessaloniki, Greece; mariasdougka@gmail.com; 11Third Department of Pediatrics, Aristotle University of Thessaloniki, Hippokration General Hospital, 546 42 Thessaloniki, Greece; ktcharisi@auth.gr; 12First Department of Pediatrics, Aristotle University of Thessaloniki, Hippokration General Hospital, 546 42 Thessaloniki, Greece; th.mpagg@gmail.com (A.B.); farmakg@auth.gr (E.F.)

**Keywords:** *Neisseria meningitidis*, invasive meningococcal disease, sequelae, outcomes

## Abstract

Invasive meningococcal disease (IMD) remains a major public health challenge due to its rapid progression, which may lead to severe sequelae or death in children and adolescents. Published data on IMD sequelae are limited in Greece and many EU countries. In the present study, patients under 16 years of age with IMD were retrospectively identified from the files of the Hellenic National Meningitis Reference Laboratory (HNML) from 2010–2020, and their medical records were tracked from the corresponding hospitals. Demographic, clinical, and microbiological data were recorded for each case. A total of 161 patients younger than 16 years of age admitted to nine hospitals across the country were identified. Of those, 91 (56.5%) records were found. The patients’ median age was 36 months (range 22 days to 16 years old); 37.4% presented with meningitis, 36.2% with both septicemia and meningitis, and 26.4% only with septicemia. The mortality rate was 5.5% and was significantly associated with septicemia, abnormal platelet count at presentation, ICU admission, and coagulation disorders, while sequelae were detected in 16.9% of patients upon discharge. *Neisseria meningitidis* serogroup B (MenB) was the most predominant (77%); of these, 269 cc was identified (36.8%). This is the first study on unfavorable sequelae and mortality due to IMD performed in Greece.

## 1. Introduction

Invasive meningococcal disease (IMD) is a severe, life-threatening disease. IMD can present as meningitis, meningitis and septicemia, or septicemia alone. Globally, approximately half a million of IMD cases are reported annually [1], with incidence rates varying across geographical regions and case fatality rates ranging from 4.1 to 20% [2]. IMD incidence rate is highest in infants and young children, followed by adolescents. In Europe, IMD is a notifiable disease, and according to recent data (2022), the annual incidence of laboratory-confirmed cases is 0.3/100,000 overall, 4.3/ 100,000 in infants under 1 year age, 0.8/100,000 in children from 1–4 years, 0.2/100,000 from the ages of 5–14 years, and 0.6/100,000 in adolescents and young adults (15–24 years) [3]. In contrast, incidence rates are considerably lower in the United States (US) (overall 0.09/100,000 in 2022), with a similar age distribution [4].

Twelve meningococcal serogroups have been identified, five of which are responsible for the vast majority of the disease burden (A, B, C, W-135, and Y) [5]. Serogroup B (MenB) is currently the most prevalent in Europe and the US; however, there is an increasing trend of serogroup W incidence (MenW) in Europe, South America, and Australia [6]. Vaccines have been developed against serogroups MenA, MenC, MenW, and MenY and recently for MenB [1,7]. In Greece, the monovalent conjugate vaccine for MenC was introduced in January 2001 and was included in the National Immunization Program (NIP) in 2005 for infants, leading to a dramatic reduction in MenC cases. In 2011, the tetravalent MCV4 vaccine was added in the NIP as a booster dose for adolescents 11–18 years of age. Furthermore, the 4CMenB, and MenB-FHbp vaccines have been available in Greece since 2014 and 2018, respectively, recommended only for high-risk groups. However, pediatricians are offering vaccination privately [8].

Internationally, the incidence of IMD varies geographically and temporally, influenced by factors such as climate, socioeconomic conditions, vaccination coverage, and serogroup distribution. Despite advancements in preventive strategies, including the introduction of meningococcal vaccines, certain regions continue to experience periodic spikes in cases. In Greece, although epidemiological data on IMD are available, including the case fatality rates [8,9], data relevant to sequelae are limited, and thus it is difficult to make informed decisions about targeting groups that would benefit most from vaccination programs.

Severe permanent sequelae in survivors consist of physical sequelae, such as severe skin necrosis, scarring requiring skin grafts and amputation of limbs, and neurological sequelae ranging from hearing loss, nerve palsy, and seizures to vegetative state, as well as psychological–behavioral disorders that include anxiety and neurodevelopmental disorders [10]. Previous studies have reported sequelae in 4.3 to 25% of IMD survivors [3,11,12].

Although surveillance of IMD in Greece is in place due to a well-constructed network that includes the National Public Health Organization (EODY), the National Meningitis Reference Laboratory, and all hospitals throughout the country, there are no reports on long-term sequelae, resulting in a gap in knowledge related to risk stratification for severe disease that could inform healthcare and public health. Furthermore, the identification of specific risk factors that lead to unfavorable sequelae or death may pinpoint patients who require more prompt and intensive interventions. The present study is the first attempt to analyze data from the IMD patients’ medical records in Greece, focusing on sequelae as well as the risk factors associated with sequelae or death.

## 2. Materials and Methods

IMD surveillance in Greece is mandatory; all cases are reported to the National Public Health Organization (EODY), while meningococcal isolates or biological fluids from suspected cases are sent to the National Meningitis Reference Laboratory (NMRL) from hospitals throughout the country by both physicians and regional microbiology laboratories.

### 2.1. IMD Case Definitions

IMD case definition was based on EU nomenclature, by which a case can be categorized as possible, probable, or confirmed based on specific clinical, laboratory, and epidemiological criteria [13]. A confirmed case is considered to be any person who meets at least one of the laboratory criteria: (i) isolation of *N. meningitidis* from a normally sterile site or purpuric skin lesions, (ii) detection of *N. meningitidis* nucleic acid from a normally sterile site or purpuric skin lesions, (iii) detection of *N. meningitidis* antigen in cerebrospinal fluid (CSF), and (iv) detection of Gram-negative diplococcus in CSF. Only confirmed cases were included in our analysis.

### 2.2. Patient Identification and Data Collection

Data collection of the laboratory-confirmed IMD cases was conducted retrospectively during an 11-year period (2010–2020), initially by retrieving the NMRL database following a manual extraction of each patient’s medical file from the participating children’s hospital or pediatric department throughout Greece. Specifically, the participation included three major children’s hospitals from the greater Athens region (“Aghia Sophia” Children’s Hospital, (AS), “P. & A. Kyriakou” Children’s Hospital, (AK), and “Pentelis” Children’s Hospital (PP)), and nine pediatric departments from six general hospitals (University General Hospital “ATTIKON”, (UGHA), University General Hospital of Alexandroupolis–Thrace (UHA), Hippokration University Hospital (Thessaloniki), “St Loukas” private hospital (Thessaloniki), Infectious Diseases Hospital (Thessaloniki), and University General Hospital of Patras, Peloponese (UHP)).

Data on demographics, clinical syndrome (meningitis and/or septicemia), hospitalization, and outcomes based on discharge status were collected and analyzed. Patients were further categorized upon admission according to the estimated Pediatric Glasgow Coma Scale (ePGCS) based upon available information such as coma followed by the need for intubation (1 point); moderately to severely altered mental status, intubation not required (2 points); mildly altered mental status (3 points); and normal mental status (4 points) as has been previously suggested [14]. Meningitis diagnoses were conducted by the participating physicians based on the medical records data. Septicemia was defined as a positive blood culture and/or positive blood PCR for *N. meningitidis* [15]. Normal ranges for white blood cells and platelet counts were adjusted for age according to standard practice [16].

Sequelae were recorded at the time of discharge, and definitions were used as described previously [11]. Sequelae were categorized as neurological, physical, and psychological–behavioral. Briefly, neurological sequelae were defined as persistent motor or sensory deficits, communication disorders, intellectual disability, abnormal brain activity (e.g., seizures), or other sever neurological disorders [11]. Seizure disorder was defined as children registered with seizures or receiving prophylactic anti-epileptic drugs upon discharge. Physical sequelae included dermatological conditions (skin scarring, skin necrosis etc.), musculoskeletal deficiencies (amputations, arthritis, etc.), cardiovascular conditions (venous thrombosis, vasculitis, etc.), renal conditions (renal failure, urinary retention, etc.), or other physical abnormalities on discharge. Psychological–behavioral sequelae included anxiety disorders, behavioral disorders, or other physiological/emotional/behavioral disorders noted at discharge.

The Glasgow Outcome Scale (GOS) [17] was retrospectively estimated for patients at the times of discharge. GOS is a disability scale of five points: death (1 point), persistent vegetative state (2 points), severe disability (3 points), moderate disability (4 points), low disability (5 points).

### 2.3. Laboratory Investigation

#### 2.3.1. Source of Specimens

A total of 161 IMD cases from children ≤ 16 years of age was confirmed during the study period from 2010–2020. All samples, cerebrospinal fluid (CSF), and/or blood (depending on the patient’s clinical presentation), as well as isolates, were sent to NMRL from hospitals throughout the country for identification and further typing.

#### 2.3.2. Identification

Meningococcal isolates were cultured on Chocolate Columbia Agar (OXOID Ltd., Basingstoke, UK) and incubated at 37 °C and 5–10% carbon dioxide for 24 h, with a further *N. meningitidis* confirmation by the application of a multiplex PCR, as previously described [18]. For the meningococcal isolates, serogroups were determined by a slide agglutination test (Remel Europe Ltd., Dartford, UK) according to the manufacturer’s instructions, while the genogroup was determined by multiplex PCR targeting specific capsule group genes, as described previously [19]. Furthermore, whenever feasible, isolates were genotyped by Multilocus Sequence Typing (MLST), and the clonal complex was determined [20,21].

### 2.4. Ethical Approval

The present study is a retrospective one, based on data from the notifiable disease registry and laboratory findings. As patient records were involved, ethical approval for conducting the study was obtained from the corresponding ethical committees of the participating hospitals.

### 2.5. Statistical Analysis

The sample size calculation was performed on the main outcome of the study, namely, the occurrence of sequelae. Assuming a proportion of approximately 17% for the overall sequelae, a sample size of 83 subjects is adequate for a ±8% range in the 95% confidence interval. The sample size calculation was performed with OpenEpi, version 3.01.

Statistical analysis was performed using the Fisher Exact Test for nominal data and the Mann–Whitney U Test for ordinal data. For proportions, 95% confidence intervals were estimated based on the binomial distribution. Results were considered statistically significant when the probability value (*p*-value) was less than 0.05.

## 3. Results

Overall, 161 confirmed cases of IMD from children aged less than 16 years old were initially identified from the NMRL records. The corresponding medical files for hospitalization were found for 91 of 161 (56.5%) patients in the nine aforementioned hospitals (Figure 1).

Demographics and clinical and microbiological data were recorded from each patient’s files. Data on fatality were available for all patients due to mandatory notification. However, data on sequelae on discharge were available for 88 of the 91 patients, while for three patients, discharge data were not available (one patient was transferred to another hospital soon after admission, and for the other two patients, the follow-up was missing as they were initially transferred to the ICU and three days later returned to a ward).

### 3.1. Data on Admission

#### 3.1.1. Age Distribution

The median age of the patients was 36 months (range: 22 days to 16 years); 40/91 (43.9%) were female. The IMD rate was higher in the age group < 1 year old, followed by 2 years old, and the rate gradually decreased in older children.

#### 3.1.2. Clinical Presentation

The median time from symptom onset to presentation in an emergency department was 24 h (range: 1.5 to 480 h). Overall, 73.6% (67/91; 95%CI: 63.3–82.3%) of children presented as meningitis. Among those, 36.2% patients also had septicemia (33/91; 95% CI: 26.4–47.0%), although septicemia was only present in 26.4% (24/91; 95%CI: 17.7–36.7%). The classic triad of meningitis (fever, nuchal rigidity, altered mental status) was present in 27.3% (24/88; 95%CI: 18.3–37.8%) of the patients. Most patients exhibited high serum inflammatory markers such as white blood cells and CRP (Table 1).

### 3.2. Laboratory Identification

Laboratory diagnosis was conducted using both PCR and culture. Among the 91 confirmed cases, 47.3% (43/91) of the cases were identified by PCR only, 35.2% (32/91) were identified by both PCR and culture, and 17.6% (16/91) of the cases were confirmed only by culture.

Serogroup B (MenB) was predominant throughout the study period, accounting for 76.9% (70/91; 95%CI: 66.9–85.1%) of the cases, followed by MenC (3%; 3/91; 95%CI: 0.7–9.3%), and MenY (1.1%; 1/91; 95%CI: 0.03–6.0%), while in 17 cases (19%) the serogroup was not identified by PCR (no culture confirmed).

MLST analysis in 54.3% (38/70) of the MenB isolates revealed that the predominant clonal complex (cc) was 269 cc (36.8%), followed by 162 cc (15.8%) and 32 cc (10.5%).

### 3.3. Outcomes

Five fatal cases were recorded during the study period, corresponding to a case fatality rate of 5.5%. Among those, the median age was 24 months (range: 5 to 108 months); four were females. Death occurred on admission day for three patients, while the remaining two fatal cases occurred on the second and third days of hospitalization. All fatal cases presented with septicemia, and one presented with meningitis and septicemia; all were admitted to the intensive care unit. Four of the fatal cases were due to MenB, while in the remaining case, the serogroup could not be identified. Notably, according to the NMRL files, the overall case fatality rate for the total 161 cases was nearly identical at 5.6% (9/161).

Sequelae upon discharge were analyzed for 83 of the 91 IMD patients. Of the remaining eight cases, three were excluded, as there were no data recorded upon discharge, and five were reported as fatal cases.

A total of 22 neurological and physical sequelae were observed in 14 patients (16.9%; 14/83; 95%CI: 9.5–26.7%) (Table 2). Among those, 11 (13.3%; 11/83) suffered only one sequela. With regard to neurological sequelae (*n* = 6), two patients suffered from balance impairment, one from paralysis, two from seizures, one from communication disorders (speech difficulties) and one from hearing loss. Physical sequelae were observed in five patients: two patients suffered from skin necrosis, two from amputation and one from skin scarring. (Table 2).

Three patients (3.6%; 3/83; 95% CI: 0.8–10.2%) suffered from a combination of five, four, and two sequelae, respectively (one patient per combination). One patient suffered five sequelae (cranial nerve palsy, muscle weakness, balance impairment, mobility problems, and seizures), one patient suffered four sequelae (paralysis, seizures, visual impairment, and hearing loss), and one patient suffered two sequelae (cranial nerve palsy and balance impairment). Among the 13 patients who had an audiogram performed during their hospitalization, hearing impairment was diagnosed in two patients. There were no documented psychological–behavioral sequelae upon discharge in our cohort.

Overall, GOS upon discharge ranged from 1–5 (median = 5) and was <5 points in 13/83 (15.6%) children. The analysis of the laboratory findings, clinical signs and syndromes (meningitis, meningitis and septicemia, septicemia), and any unfavorable sequelae or death is shown in Table 3. Impaired consciousness (EPGS < 4) at presentation was more common among patients with any sequelae (9/14, 64.3% vs. 31/66, 47%), as well as among the fatal cases (5/5 100% vs. 40/80, 50%); however, this association was found to be of no statistical significance. Mortality was higher in patients with septicemia (80% (4/5), vs. 18% (15/83, *p* = 0.007). Patients who received plasma for coagulation disorders were more likely to suffer from unfavorable sequelae and death. ICU admission and low platelet count at presentation were significantly associated with mortality.

## 4. Discussion

This is the first study investigating clinical characteristics and outcomes due to IMD in the pediatric population of Greece. According to the data presented, the overall rate of sequelae upon discharge was 16.9%, and the case fatality rate was 5.5%. Septicemia, low platelet count at presentation, ICU admission, and coagulation disorders that required plasma administration were all associated with mortality. MenB was the most predominant (77%) serogroup, and among these, 269 cc and 32 cc were the most predominant clonal complexes during the study period. These clonal complexes belong to the hyperinvasive clones and have been commonly associated with disease [22].

Although there are some recent publications on IMD from Greece [8,23,24], no data have been reported on sequelae. The predominant serogroup in our study was MenB (77%), and the incidence fluctuated between 62% (2016) and 100% (2017) per year. The predominance of MenB in Greece as has also been reported previously [8,23,24]; are in agreement with published European data [25] and are probably related to some extent to the inclusion of the monovalent (Men C) in children over one year of age and the quadrivalent (Men ACWY) vaccines in adolescents over 11 years of age in the National Immunization Program in 2005 and 2011, respectively. Notably, in cases for which the information was available, none of the children was vaccinated against MenB. Interestingly, a decline in IMD from the MenB serogroup was observed in some European countries (e.g., Ireland, UK) during the same period, which may be partly explained by the implementation of MenB vaccination in the respective national immunization programs [26]. In other European countries (e.g., Poland), case numbers of MenB did not decline over time, resembling our findings [25,26].

This is the first study exploring outcomes of IMD in Greece. In light of the emerging preventive IMD strategies such as effective vaccines for MenB [1,27], knowledge of the sequelae rate following IMD is a sine qua non for estimating the disease burden. In our cohort, 16.9% of patients were discharged with complications, which is in agreement with previously published data on similar cohorts reporting a rate of sequelae ranging from 15 to 30% among IMD survivors [12,28,29,30,31,32,33].

Neurological complications were present in 10.8% of the patients, which is in line with previously published studies that reported neurological complications ranging from 5.5 to 21% [28,31,33]. Hearing impairment was present in two patients (2.4%) in our study, while other reports show that hearing impairment among IMD survivors ranges from 2.6% to 21% [31,34,35]. The low rate of hearing impairment in our study may be attributed to the relatively limited audiological screening before discharge, which was provided to only 13 patients. This emphasizes the necessity for the audiology testing of all patients with IMD—especially with meningitis—to detect any subtle hearing impairment.

Physical sequelae, dermatological conditions in particular, were observed in 6% of patients in the present study. Dermatological conditions, especially skin necrosis–scarring and amputation, may be debilitating for survivors and were observed in 3.6% and 2.4%, respectively. This is in accordance with previously published data, which report rates of skin necrosis–scarring ranging from 2.4 to 10.1% [2,36] and rates of amputation ranging from 2.7 to 7.6% [31,33].

In most cases, sequelae are evaluated after discharge and may include psychological as well as neurodevelopmental disorders, deficits in executive function, and multiple aspects of memory problems. These neurological and cognitive issues, as well as psychological and behavioral/developmental outcomes, should be evaluated after discharge and preferably via long-term follow-up to be accurately characterized. Lack of psychological and behavioral sequelae in our cohort may have contributed to the overall percentage of sequelae, as long-term follow-up of the patients was not recorded.

A study focusing on psychological disorders after 12-month follow-ups reported that 11% of survivors were at risk for post-traumatic stress disorder (PTSD) [37], while a recent study added the autism spectrum of disorders as a possible IMD sequela [33].

Furthermore, it is recognized that the sequelae of IMD can be lifelong, and patients can have multiple sequelae involving different body systems, resulting in devastating impacts on health-related quality of life among the survivors and their caregivers [38]. Thus, lack of data from follow-ups of IMD survivors may cause underestimation of the burden of IMD in Greece. Consequently, in order to guide better public health policies, especially with regard to vaccination for IMD, research on the long-term sequelae of IMD should be prioritized.

The case fatality rate of 5.5% in the present study is in accordance with previous studies that reported case fatality rates from 4.1 to 20% [2]. In a recent review and meta-analysis by Wang et al., the pooled case fatality ratio of IMD due to MenB was reported to be 6.9% (95%CI: 6.0–7.8%); this is lower than other serogroups and is similar to the case fatality ratio in our study [2].

Knowledge of specific clinical and laboratory markers that are associated with serious outcomes may identify patients who may benefit from more aggressive interventions. Many risk factors have been correlated with the development of complications or death due to IMD. These include older age, septicemia, purpura fulminans [39], septic shock, ICU admission, and short time from symptom onset [31]. In accordance with previous studies, septicemia, low platelet count on admission, ICU admission, and coagulation disorders that require plasma administration were all associated with mortality in the current cohort. Thrombocytopenia may accompany sepsis via multiple mechanisms, which include decreased production, interaction among platelet receptors, immune-mediated sequestration, consumption, etc., and has been associated with mortality in multiple settings [40] but especially in meningococcal sepsis [41]. Lack of association between mortality and older age, as has been suggested, is probably due to the fact that the present study was carried out on the pediatric population, whereas an association of IMD with mortality was found by other researchers [39] in patients over 50 years of age.

The main strength of our study is that it reviews data on pediatric patients with IMD from most parts of Greece, not just the area of Athens. However, as the study was retrospective, incompleteness of data and non-standardized protocols in diagnosis and treatment were inevitably observed. Furthermore, due to lack of common definitions concerning sequelae, the outcomes were not directly comparable to other studies, a very important issue that must be addressed by specialists in the field. Another limitation is that there were no data on follow-ups after hospital discharge of IMD survivors. Many sequelae could have improved or worsened after hospital discharge, and this should be accounted for when analyzing the results of the present study. Additionally, as no data or limited data are provided on behavioral or physiological disorders, further studies focusing on the above data are essential to better understand this aspect and for the patients to receive further appropriate medical care. Moreover, no multivariate analysis was undertaken, as this was beyond the scope of our study, although this would provide a more descriptive overview of invasive meningococcal disease in Greek children.

Finally, only 56.5% of relevant medical records and files used in this study were complete. If medical records are kept only as paper copies, updates need to be carried out manually, as having no backup or alternate storage makes them prone to damage and loss [42]. However, the similar fatality rate between the confirmed 161 cases in the NMRL records and the 91 children whose medical records were retrieved and analyzed in this study (5.6% and 5.5%, respectively) favors the representative nature of the results in the present study. Nonetheless, this limitation underlines the necessity of digitalizing all medical records in Greece.

## 5. Conclusions

In conclusion, the present study demonstrated that unfavorable sequelae were observed in 16.9% of the included patients and case fatality was 5.6% in children with IMD in Greece. More uniform definitions of sequelae are required to enable more direct comparisons. Septicemia, abnormal platelet count upon admission, ICU admission, and coagulation disorders requiring plasma administration were all associated with mortality.

## Figures and Tables

**Figure 1 microorganisms-13-00705-f001:**
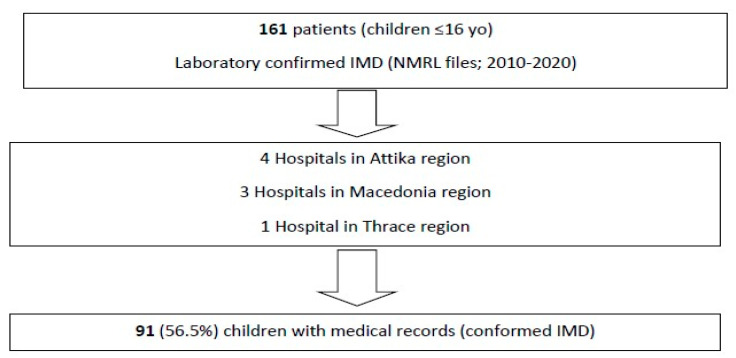
Flow diagram of this study’s process.

**Table 1 microorganisms-13-00705-t001:** Clinical and laboratory data on the admission of patients with IMD in Greece from 2010–2020 (*n* = 91).

Age (*n* = 91)	Median = 36 Months (Range: 22 Days–16 Years)
Sex (*n* = 91)	43.9% (40/91) female
Time from symptom onset (*n* = 86/91)	median = 24 h (range: 1.5–480 h)
Meningitis triad (*n* = 88/91)	27.3% (24/88)
ePGCS (*n* = 88/91)	median = 3 (range 1–4)
ePGCS < 4	51.1% (45/88)
White blood cells (/μL) (*n* = 79/91)	median = 15,140 (range: 1509–43,900)
Platelets (/μL) (*n* = 79/91)	median = 243,000 (range: 18,000–769,000)
C-reactive protein (CRP) (mg/L) (*n* = 77/91)	median = 92.96 (range: 0–515)
Creatinine (mg/dL) (*n* = 77/91)	median = 0.5 (range: 0.13–2.5)
Glucose (mg/dL) in serum (*n* = 78/91)	median = 117.5 (range: 10–352)
CSF white blood cells (/μL) (*n* = 65/91)	median = 2000 (range: 0–28,000)
CSF glucose (mg/dL) (*n* = 61/91)	median = 40 (range: 0–147)
CSF protein (mg/dL) (*n* = 59/91)	median = 91.4 (range: 10–636)

Abbreviations: ePGCS: Pediatric Glasgow Coma Scale; CSF: cerebrospinal fluid.

**Table 2 microorganisms-13-00705-t002:** Sequelae among the 14 children with IMD upon discharge (2010–2020) in Greece.

Patients with Sequelae			Single Sequelae (*n* = 11)	Multiple Sequelae (*n* = 3)	Overall Sequelae (*n* = 22)
**NEUROLOGICAL** **SEQUELAE**	**Motor deficits**	Cranial nerve palsy	0	2 (patients 1 and 2)	2 (9%)
		Muscle weakness	0	1 (patient 1)	1 (4.5%)
		Balance impairment	2	2 (patients 1 and 2)	4 (18.2%)
		Mobility problems	0	1 (patient 1)	1 (4.5%)
		Paralysis	0	1 (patient 3)	1 (4.5%)
	Abnormal brain activity	Seizures	2	2 (patient 1 and 3)	4 (18.2%)
	Communication disorders	Speech difficulties	1		1 (4.5%)
	Sensory deficits	Visual impairment	0	1 (patient 3)	1 (4.5%)
		Hearing loss	1	1 (patient 3)	2 (9%)
**PHYSICAL** **SEQUELAE**	Dermatological conditions	Skin necrosis	2		2 (9%)
		Amputation	2		2 (9%)
		Skin scarring	1		1 (4.5%)
				Patient 1 (*n* = 5) Patient 2 (*n* = 2) Patient 3 (*n* = 4)	
Total number of patients			**11**	**3**	

**Table 3 microorganisms-13-00705-t003:** Correlation of clinical symptoms and signs, laboratory values, and sequelae upon discharge or death in 88 children (the three children who were lost to follow up were excluded).

	Children with Sequelae (*n* = 14)	Children Without Sequelae (*n* = 69)	¥	Fatal Cases (*n* = 5)	Survived (*n* = 83)	*p* Value ^¥^
Age (median)	8 mo–8 yo (24 mο)	22 d–16 yo (39 mο)	0.62	5 mo–9 yo (24 mo)	22 d–16 yo (36 mo)	0.92
EPGS < 4	9/14 (64.3%)	31/66 (47%)	0.37	5/5 (100%)	40/80 (50%)	0.057
Meningitis triad	3/14 (21.4%)	21/66 (31.2%)	0.535	0/5	24/80 (30%)	0.31
Meningitis	5/14 (55.5%)	33/69 (47.8%)	0.56	0/5	38/83 (4.7%)	0.067
Meningitis and septicemia	7/14 (50%)	23/69 (33.3%)	0.36	1/5 (20%)	30/83 (36.1%)	0.64
Septicemia	2/14 (14.3%)	13/69 (18.8%)	1	4/5 (80%)	15/83 (18.1%)	0.007
Abnormal white blood cells upon admission	5/11 (45.5%)	37/60 (61.7%)	0.5338	2/5 (40%)	42/71 (59.2%)	0.65
Abnormal (low) platelet count on admission	2/11 (18.2%)	6/60 (10%)	0.6	3/5 (60%)	7/71 (16.9%)	0.0147
ICU admission	6/14 (42.8%)	21/66 (31.8%)	0.53	5/5 (100%)	27/80 (33.75%)	0.006
Plasma administration	5/14 (35.7%)	14/66 (21.2%)	0.76	4/5 (80%)	19/80 (23.75%)	0.0005

^¥^ Statistical hypothesis was tested by the Fisher Exact or Mann–Whitney U Tests.

## Data Availability

All data are available within the article.

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
