# Peer review of "Invasive Meningococcal Disease in Children: Outcomes and Risk Factors for Sequelae and Fatal Cases in Greece"

_microorganisms, 2025, doi:10.3390/microorganisms13040705_

Round 1
Reviewer 1 Report
Comments and Suggestions for Authors
General comments
In their manuscript, Dr. Panagiotis Poulikakos and colleagues described IMD in pediatric patient population in Greece. They have provided valuable clinical information albeit there are some deficiencies that I suggest the authors to carefully consider and revise accordingly before the manuscript can be deemed suitable for publication.
Specific comments
- In the abstract. lines 38-40: “ A total of 9 hospitals across the country participated in the present study aiming to analyze 161 patients admitted with IMD younger than 16 years of age according to the HNML records.” Should this read as “A total of …. aiming to analyze 161 patients younger than 16 years of age admitted with IMD according to ….. records”.
- In Figure 1, the word “conformed” should be “confirmed”? Also Figure 1 contains numbers which I assume are line numbers? They should be remove from the figure.
- Abbreviations for Table 1 should be defined as footnotes.
- Results Section, 3.2: lines 222-223: while in 17 cases (19%) the serogroup was not identified. Please indicate if these were PCR diagnosed cases without bacteriological culture?
- Results Section 3.2: lines 219-220, the authors stated that “35.2% (32/91) by both PCR and culture, while only 17.6% (16/91) of the cases were confirmed only by culture.”. This means there are a total of 48 cases with positive bacterial cultures. Why then in lines 224, the authors stated that “MLST analysis in 80% (56/70) of the MenB isolates)” which suggested there were 70 MenB isolates?
- Results section 3.3, lines 227-229, “case fatality rate of 5.5%”; please add number of fatal cases since it was unclear what was the denominator to give 5.5%.
- Table 2, in the last row for Total number of patients, and under the far right column, what is meant by patient 1 (n = 5)? Does it mean this single patient had five sequelae? In the title, the authors indicated that there were 14 children with sequelae but in the last row of table, the total number of patients was 11?
I suggest to revise table 2 to list the 14 children as patient 1 to patient 14 and describe their sequelae please. If this is done then the descriptions on lines 247-254 could be simplified to summarize the results in one or two sentences.
Author Response
RESPONSE to Reviewer-1
Comment-1: In the abstract. lines 38-40: “ A total of 9 hospitals across the country
participated in the present study aiming to analyze 161 patients admitted with IMD
younger than 16 years of age according to the HNML records.” Should this read as “A
total of …. aiming to analyze 161 patients younger than 16 years of age admitted with
IMD according to ….. records”.
Authors reply: Thank you for this comment. The manuscript underwent an English
Language review by a native speaker. The sentence was already rephrased. However,
the age of the patients was also added according to your comment.
Comment-2: In Figure 1, the word “conformed” should be “confirmed”? Also Figure 1
contains numbers which I assume are line numbers? They should be remove from the
figure.
Authors reply: thank you for your comments. The word is corrected and the numbers
are removed from the table.
Comment-3: Abbreviations for Table 1 should be defined as footnotes
Authors reply: abbreviations are added in Table 1
Comment-4: Results Section, 3.2: lines 222-223: while in 17 cases (19%) the serogroup
was not identified. Please indicate if these were PCR diagnosed cases without
bacteriological culture?
Authors reply: Thank you for the comment. Indeed, the 17 cases were confirmed
by PCR only. In the Results Section, 3.2 the following sentence has been added
…while in 17 cases (19%) the serogroup was not identified by PCR (no culture
confirmed).
Comment-4: Results Section 3.2: lines 219-220, the authors stated that “35.2% (32/91) by
both PCR and culture, while only 17.6% (16/91) of the cases were confirmed only by culture.”.
This means there are a total of 48 cases with positive bacterial cultures. Why then in lines
224, the authors stated that “MLST analysis in 80% (56/70) of the MenB isolates)” which
suggested there were 70 MenB isolates?
Authors reply: Thank you for this very useful comment. Reviewing the data
once more, it was found that a total of 38 culture confirmed cases underwent MLST
analysis all were MenB. The numbers are corrected at the text Results Section 3.2
last paragraph as : “…MLST analysis in 54.3% (38/70) of the MenB isolates revealed
that the predominant clonal complex (cc) was 269cc (36.8%), followed by 162 cc (15.8%)
and 32cc (10.5%)” as well as in the abstract.
Comment-5: Results section 3.3, lines 227-229, “case fatality rate of 5.5%”; please add
number of fatal cases since it was unclear what was the denominator to give 5.5%.
Authors reply: Thank you for this comment. The number is also added. The rate is
corrected to 5.6% instead of 5.5%
Comment-6: Table 2, in the last row for Total number of patients, and under the far
right column, what is meant by patient 1 (n = 5)? Does it mean this single patient had
five sequelae? In the title, the authors indicated that there were 14 children with
sequelae but in the last row of table, the total number of patients was 11?
Authors reply: Thank you for this clarification. Indeed the total number of 14
patients suffering sequelae. According to Table 2, 11 patients suffered from single
sequelae, while, 3 patients suffered from multiple sequelae (total number 14).
Indeed, one patient suffered from 5 sequelae.
Comment-7: I suggest to revise table 2 to list the 14 children as patient 1 to patient 14
and describe their sequelae please. If this is done then the descriptions on lines 247-
254 could be simplified to summarize the results in one or two sentences.
Authors reply: Thank you for this comment. We have tried various tables before
the submission. However, we found that Table 2 was the most simplified solution of
showing both Neurological and Physical sequelae at a glance.
Reviewer 2 Report
Comments and Suggestions for Authors
The study investigates the long-term sequelae and risk factors associated with invasive meningococcal disease (IMD) in Greece, addressing a significant gap in epidemiological data. By retrospectively analyzing patient records from the Hellenic National Meningitis Reference Laboratory (HNML) and corresponding hospitals, the study included cases admitted between 2010 and 2020. The research focused on 161 pediatric patients under 16 years of age, with 91 complete records analyzed. The study highlights the severity of IMD, with a fatality rate of 5.5% and sequelae observed in 16.9% of patients at discharge. The findings emphasize the critical need for early diagnosis and management to mitigate the disease's long-term consequences.
The results indicate that septicemia, abnormal platelet counts, ICU admission, and coagulation disorders were significantly associated with increased mortality. The predominant pathogen identified was Neisseria meningitidis serogroup B (MenB), accounting for 77% of cases, with a notable presence of the 269cc strain (33.9%). The study underscores the clinical burden of IMD, as nearly two-thirds of cases presented with meningitis alone or in combination with septicemia. These findings align with global trends, reinforcing the necessity of comprehensive surveillance and targeted vaccination strategies to prevent severe outcomes.
As the first study to assess IMD-related sequelae and mortality in Greece, this research provides crucial insights into the disease's impact on pediatric populations. The study's retrospective design and limited dataset pose constraints, yet the results serve as an important reference for public health strategies. Further research is essential to expand the dataset and explore additional risk factors influencing IMD outcomes. Strengthening vaccination programs and improving early detection measures could play a pivotal role in reducing the disease burden and improving patient prognosis.
I make some suggestions below:
- Perform sample size calculation
- Figure 1 is out of format, adjust it.
- The graph presented in Figure 2 could be improved aesthetically. However, I believe it is not appropriate, and the data should be presented descriptively in the text of the article.
- Table 1 is very cluttered, I suggest readjusting it and making it cleaner.
- Table 3 also needs adjustments, with the aim of making it clearer. For example, in the p-value column, the letter p does not need to appear in all rows.
- The analyses could be more detailed, performing a multivariate analysis, with the aim of establishing independent associations. In addition, it would be interesting to present an effect measure with 95% CI.
Author Response
RESPONSE to Reviewer 2
Comment-1: Perform sample size calculation
Authors reply: Thank you for this comment that allowed us to provide with additional details regarding the background of our study design. The relevant passage reads: “The sample size calculation was performed on the main outcome of the study, namely the occurrence of sequelae. Assuming a proportion of 17% approximately for the overall sequelae, a sample size of 83 subjects is adequate for a +/-8% range in the 95% confidence interval. The sample size calculation was performed with OpenEpi version 3.01.” (page 4, section 2.5 Statistical analyses.)
Comment-2: Figure 1 is out of format, adjust it.
Authors reply: Thank you for this comment. This has been addressed accordingly in the manuscript (page 5, section 3. Results, Figure1 )
Comment-3: The graph presented in Figure 2 could be improved aesthetically. However, I believe it is not appropriate, and the data should be presented descriptively in the text of the article.
Authors reply: Thank you for this comment. We have removed the figure and described the findings in the manuscript (page 5, section 3.1.1 Age distribution)
Comment-4: Table 1 is very cluttered, I suggest readjusting it and making it cleaner.
Authors reply: Thank you for this comment. This has been addressed accordingly in the manuscript : (page 6, section 3.1.2. Clinical presentation, Table 1)
Comment-5: Table 3 also needs adjustments, with the aim of making it clearer. For example, in the p-value column, the letter p does not need to appear in all rows.
Authors reply: Thank you for this comment. This has been addressed accordingly in the manuscript : (page 8, section 3.3. Outcomes, Table 3)
Comment-6: The analyses could be more detailed, performing a multivariate analysis, with the aim of establishing independent associations. In addition, it would be interesting to present an effect measure with 95% CI.
Authors reply: Thank you for this enlightening comment; we added 95% CI in the major proportions throughout the manuscript. The 95% Cis were estimated assuming a binomial distribution; this has been declared in the revised Methods (page 4 paragraph 2.5) . On the other hand, we have not performed a multivariate analysis, as this was beyond the scope of our study providing a more descriptive overview of Invasive Meningococcal Disease in Greek children. We declared the lack of multivariate analysis as a limitation of the study, to present objectively your kind remark to the audience.(page 10, section 4. Discussion, paragraph 11, lines 365-367)
Reviewer 3 Report
Comments and Suggestions for Authors
- The public health implications of the absence of follow-up data in Greece must be elucidated.
- The recommendation is for additional peer review by statisticians to validate test selection and enhance reporting standards.
- The types of sequelae and the numbers/percentages should be listed in Table 2.
- The discussion should be expanded to compare MenB infection rates in Greece with serogroup changes after vaccination in other European countries.
- The clinical significance of low platelet counts as a risk factor for death should be discussed.
- The lack of assessment of long-term sequelae, such as possible psychological/neurodevelopmental consequences after hospital discharge, is also a matter for concern.
It would be better, if authors could review the whole manuscript, as some grammatical mistakes were noted in context.
Author Response
RESPONSE to Reviewer 3
Comment-1: The public health implications of the absence of follow-up data in Greece must be elucidated.
Authors reply: Thank you for this valuable comment. We may have not clarified the impact of the absence of follow up data. This has been addressed in the manuscript (page 9, section 4. Discussion, paragraph 8, lines 328-333)
Comment-2: The recommendation is for additional peer review by statisticians to validate test selection and enhance reporting standards.
Authors reply: Thank you for this comment. Your suggestion has been addressed in the manuscript which underwent statistical analysis (page 4, section 2.5, statistical analysis)
Comment-3 :The types of sequelae and the numbers/percentages should be listed in Table 2.
Authors reply: Thank you for this helpful comment. This has been addressed in the manuscript (page 7, section 3.3. Outcomes, Table 2)
Comment-4 The discussion should be expanded to compare MenB infection rates in Greece with serogroup changes after vaccination in other European countries.
Authors reply: Thank you for this enlightening comment. This has been addressed in the manuscript (page 9, section 4. Discussion, paragraph 2 lines 285-297)
Comment-5 The clinical significance of low platelet counts as a risk factor for death should be discussed.
Authors reply: Thank you for this comment. This has been addressed in the manuscript (page 10, section 4 Discussion, paragraph 10)
Comment-6 The lack of assessment of long-term sequelae, such as possible psychological/neurodevelopmental consequences after hospital discharge, is also a matter for concern.
Authors reply Thank you for this comment. Indeed, we believe that long term sequelae should be assessed and this is highlighted in the discussion. Nonetheless, presented in the manuscript are the best available data on sequelae in IMD survivors in Greece and we hope that this publication may act as a trigger for future long term surveillance of IMD survivors for psychological/neurodevelopmental consequences (page 9, section 4. Discussion, paragraph 8 lines 328-333)
Comment-7 Comments on the Quality of English Language. It would be better, if authors could review the whole manuscript, as some grammatical mistakes were noted in context.
Authors reply Thank you for this comment. This has been addressed throughout the manuscript.
Reviewer 4 Report
Comments and Suggestions for Authors
The Author's described IMD pediatric cases in Greece from 2010 to 2029. In particular that focus on sequelae of the IMD. The study is interesting and well written.
However some improvements could by necessary. In particular I suggest:
- do the authors have any informations about post-discharge sequelae? Some of the clinical issues described in discharge could improved (or even worsened) during a relatively short follow up period after discharge and this information could be very interesting for the reader
- - please add a paragraph about the limitations of the study talking also about the lack of follow up
- - the era some typos in the text, please correct them
Author Response
Comment-1 do the authors have any informations about post-discharge sequelae? Some of the clinical issues described in discharge could improved (or even worsened) during a relatively short follow up period after discharge and this information could be very interesting for the reader
Authors reply Thank you for this insightful comment. Unfortunately, there are no information about post-discharge sequelae in Greece. Nonetheless, presented in the manuscript are the best available data on sequelae in IMD survivors in Greece and we hope that this publication may act as a trigger for future long term surveillance studies of IMD survivors. (page 9, section 4. Discussion, paragraph 8 lines 328-333)
Comment-2. please add a paragraph about the limitations of the study talking also about the lack of follow up
Authors reply Thank you for this comment. This has been addressed in the manuscript in the discussion (page 9 and 10 section 4. Discussion, paragraph 8 lines 328-333 and paragraph 11, lines 354-367
Comment-3. The era some typos in the text, please correct them
Authors reply Thank you for this comment. This has been addressed throughout the manuscript.
Round 2
Reviewer 2 Report
Comments and Suggestions for Authors
The authors have performed the changes as recommended. The manuscript can be accepted for publication.